# Hypnosis Sedation Used in Breast Oncologic Surgery Significantly Decreases Perioperative Inflammatory Reaction

**DOI:** 10.3390/cancers17010049

**Published:** 2024-12-27

**Authors:** Martine Berliere, Fabienne Roelants, François P. Duhoux, Amandine Gerday, Nathan Piette, Camille Lacroix, Marie-Agnes Docquier, Vasiliki Samartzi, Maude Coyette, Jennifer Hammer, Nassim Touil, Houda Azzouzi, Philippe Piette, Christine Watremez

**Affiliations:** 1Breast Clinic, King Albert II Cancer Institute, Clniques Universitaires Saint Luc, Université Catholique de Louvain, 1200 Woluwé-Saint-Lambert, Belgium; francois.duhoux@saintluc.uclouvain.be (F.P.D.); amandine.gerday@saintluc.uclouvain.be (A.G.); maude.coyette@saintluc.uclouvain.be (M.C.); jennifer.hammer@saintluc.uclouvain.be (J.H.); houda.azzouzi@student.uclouvain.be (H.A.); 2Department of Anesthesiology, Cliniiques Universitaires Saint Luc, Université Catholique de Louvain, 1200 Woluwé Saint-Lambert, Belgium; fabienne.roelants@saintluc.uclouvain.be (F.R.); marie-agnes.docquier@saintluc.uclouvain.be (M.-A.D.); nassim.touil@saintluc.uclouvain.be (N.T.); christine.watremez@saintluc.uclouvain.be (C.W.); 3Department of Pneumology, CHU UCL Namur, 5530 Yvoir, Belgium; nathan.piette@student.uclouvain.be; 4Department of Gynaecology, CHU UCL Namur, 5000 Namur, Belgium; camille.lacroix@chuuclnamur.be; 5Department of Gynaecology, Hopital de Jolimont, 7100 La Louvière, Belgium; vasiliki.samartzi@hopitaldejolimont.be; 6Medical and Financial Department, Grand Hopital de Charleroi, 6000 Charleroi, Belgium; philippe.piette@ghdc.be

**Keywords:** breast cancer, oncologic surgery, hypnosis sedation, VRH (virtual reality hypnosis), CRP (C-reactive protein), NLR (neutrophil-to-lymphocyte ratio), anti-inflammatory mechanisms

## Abstract

Hypnosis sedation has been used for anesthesia in breast oncologic surgery. This manuscript reports the results of a multicentric, prospective non-randomized study evaluating three different modalities of anesthesia for breast cancer surgery: general anesthesia, general anesthesia preceded by virtual reality with a hypnorelaxation session and hypnosis sedation exclusively in place of general anesthesia. Local anesthesia was systematically administered in all patients. Some benefits of hypnosis sedation, a decrease in pain and the consumption of non-steroidal anti-inflammatory drugs are correlated with a significant reduction in the inflammatory reaction in the perioperative process.

## 1. Introduction

For breast cancer patients, the latest decade has been marked by therapeutic progress and a decrease in mortality. These remarkable advances are due to scientific progress and a multidisciplinary approach. These observations are very encouraging, but breast cancer treatments are still associated with important side effects, which are deleterious for patients’ quality of life [1,2,3,4,5,6,7,8].

In this context, one of the major challenges for clinicians is to reduce the side effects of anti-cancer treatments.

Hypnosis sedation has already demonstrated benefits in different modalities of breast cancer treatment, particularly in breast cancer surgery to avoid general anesthesia [9,10,11,12,13,14]. However, in an attempt to prove the reproducibility of these observed benefits and to try to explain the implicated mechanisms, studies with an evaluation of different biological parameters need to be performed.

This study focuses on inflammatory parameters and compares the evolution of these parameters among patients operated on while on hypnosis sedation or operated on while on general anesthesia or general anesthesia preceded by a virtual reality hypnosis session.

Virtual reality hypnosis allows complete immersion in an artificial three-dimensional environment (where the patient is isolated from the outside world) with music and a hypnosis relaxation session. Unlike hypnosedation, virtual reality with hypnosis does not require the presence of an anesthetist trained in hypnosis.

The hypothesis sustained by our study is that the benefits induced by hypnosis sedation can partially be explained by a significant decrease in the perioperative inflammatory reaction.

Moreover, are these benefits the result of a personalized hypnotic approach, or is a hypnorelaxation session combined with virtual reality sufficient to generate benefits?

## 2. Materials and Methods

Between January 2017 and October 2019, 284 patients from the Breast Clinic of the King Albert II Cancer Institute (Clinique Universitaires Saint-Luc, Université Catholique de Louvain, Brussels, Belgium) and from Jolimont Hospital La Louviere, Belgium were included in a prospective interventional non-randomized study approved by our two local ethics committees (clinicaltrials.gov: NCT03330177).

Informed consent was obtained for all patients. Four patients were excluded (a withdrawal of informed consent for one patient and major discomfort during the hypnorelaxation session with discontinuation of this session for three patients).

Ninety-three patients underwent breast surgery (lumpectomy or mastectomy + axillary lymph node dissection or sentinel lymph node biopsy) while on general anesthesia (group GA).

Ninety-two patients underwent the same surgical procedures while on general anesthesia preceded by virtual reality with a hypnorelaxation session (AQUA Oncomfort/HypnoVRTM) (group GAVRH).

Finally, 95 patients underwent the same procedures while on hypnosis sedation (group HYPS).

### 2.1. Description of Hypnosis Sedation Procedure, General Anesthesia and Virtual Reality with Hypnorelaxation Session

Patients were evaluated before surgery during a preoperative anesthesiology consultation. In the HYPS group, patients received specific explanations about hypnosis sedation. During this session, the modalities and the course of the procedure were described to patients, and physicians confirmed that they were adequate candidates for this kind of analgesia and anesthetic procedure, i.e., that they were able to sign an informed consent form and able to understand the languages spoken at our institution. No patient requesting hypnosis sedation was refused. One hour before surgery, premedication with oral lorazepam (0.5 mg) was proposed to the patient. At the time of the surgical procedure, all the patients were monitored classically (electrocardiography, noninvasive blood pressure measurement, blood oxygen saturation assessment [SpO_2_] and capnography). Local anesthesia was performed with a combination of levobupivacaine 0.25% and lidocaine 0.5%. Oxygen was administered to each patient. Once the patient was comfortably positioned on the operating table, the anesthesiologist induced hypnosis as a procedure in which indirect suggestions were given based on the anesthesiologist’s observation of the patient’s behavior and on her or his judgement of the patient’s needs. The patients were invited to fix their gaze on a point in front of them while concentrating on their body to achieve total muscle relaxation before finally closing their eyes. Guided by the anesthesiologist, the patients had to focus their attention on a positive recollection. By using a calm and monotonous voice, the anesthesiologist constantly talked to help them relive a dream or experience so that they remained as detached and dissociated as possible from the reality surrounding them. A state of intense well-being and comfort had to be reached and maintained during the whole procedure. The peri-incisional skin was injected with a local anesthetic such as 0.5% lidocaine combined with 0.25% levobupivacaine. A continuous infusion of remifentanil, a very short-acting µ-opioid agonist, was started at a rate of 0.05 µg/kg/min (a dose about 10 times lower than the one used for general anesthesia) and was modified or stopped as required. If an anxiolytic was needed, small doses of midazolam were administered through the IV line 0.1 mg at a time. A pre-established communication system between the anesthesiologist and their patients allowed the latter to express any discomfort. In such a case, the hypnotic state was strengthened, the surgeon could improve local anesthesia or the infusion rate of remifentanil could be increased. Once the procedure was completed, the anesthesiologist gave the patients recommendations (posthypnotic suggestions) in order to preserve their comfort in the postoperative period, to promote correct healing, to keep the wound dry and to give the patient the opportunity to reuse hypnosis during their cancer treatment.

None of the patients in the HYPS group included in the current study required a conversion to general anesthesia. In this group, patients thus maintained consciousness during the whole surgical procedure and avoided a pharmacological coma. General anesthesia was administered following the usual institutional procedures, based on interventional guidelines.

Premedication with oral lorazepam was the same in the 3 groups. In an attempt to reduce bias, pre- and postoperative suggestions were given by the anesthesiologists to patients undergoing surgery while on general anesthesia. Local anesthesia was injected before skin incision. General anesthesia was induced by the intravenous administration (continuous infusion) of propofol (2–3 mg/kg), with IV lidocaine hydrochloride (1 mg/kg) ketamine hydrochloride 0.3 mg/kg, and sufentanil 0.1–0.2 µg/kg and cis-atracurium if necessary. The airway was secured with an endotracheal or supraglottic tube, and the lungs were ventilated with a mixture of oxygen and air (50–50%); the tidal volume was set at 6–8 mL/kg ideal body weight. Anesthesia was maintained with the intravenous administration of propofol (target-controlled infusion devices), and additional intravenous sufentanil citrate (5 µg) was administered during surgery if the heart rate or surgical blood pressure increased by more than 20%. In the days after surgery, pain was controlled following the institution’s protocol: paracetamol 1 g/6 h and naproxen 500 mg/12 h in the case of low pain, tramadol 50 mg/6 h in the case of mild pain and piritramide 20 mg/12 h in the case of severe pain. These medicines were given to patients as required.

In the GAVRH group, patients received a preoperative virtual reality session with hypnorelaxation, and the program was administered just before they entered the operating room. This Oncomfort program is named AQUA and corresponds to immersion in an aquatic environment. The AQUA VRH session follows the standard phases of hypnosis. Patients were invited to focus their attention on their breath to induce progressive relaxation. The guidance phase brought the subject slowly under the water. During the deepening phase, patients followed an underwater experience with specific suggestions regarding their comfort and relaxation. The re-altering phase brought the patients progressively back to reality with a return to normal body sensations. The short duration of the session (20 min) is probably responsible for the absence of observations of a positive impact on side effects.

### 2.2. Parameter Evaluation

Clinical parameters such as pain score, anxiety and distress score were measured on day 0 (before the VRH session), day 1 and day 8. Consumption of non-steroidal anti-inflammatory drugs (NSAIDs) was recorded on day 0, day 1 and day 8. These evaluations were performed independently by nurses and physicians. Patients’ distress severity was measured with the National Comprehensive Cancer Network distress thermometer. This is a visual analog scale in the form of a thermometer. A score equal to or greater than 5 on the scale should draw attention, and patients should be referred to psychological services. Anxiety was assessed by the VAS-A (visual anxiety scale) [15].

Pain intensity was measured by visual analog scales on days 0, 1 and 8, resulting in a numerical pain rating score (NRS) [16].

We studied the following biological parameters: neutrophil-to-lymphocyte ratio (NLR) and C-reactive protein (CRP). A sub-study measured different endocannabinoids but is not described in this paper.

CRP is an acute-phase protein which represents a very sensitive marker of inflammatory and tissue damage. Elevated CRP is common in infection, tissue damage and acute and chronic inflammatory diseases.

In peripheral blood, variations in CRP have been associated with the progression of breast cancer [17,18].

NLR (neutrophil-to-lymphocyte ratio): The NLR is calculated by dividing neutrophils into lymphocytes, which reflect neutrophilia or lymphopenia. Inflammation is thought to play a key role in carcinogenesis and cancer progression because tumors cause neutrophilia by upregulating the granulocyte colony-stimulating factor (G-CSF) and the granulocyte-macrophage colony-stimulating factor (GM-CSF). Neutrophils promote tumor growth through different mechanisms such as the stimulation of angiogenesis, the inhibition of apoptosis and DNA damage [19].

Their production and migration may suppress the cytolytic activity of other cells such as lymphocytes, which also play an important role in host tumor immunity. Lymphopenia means that the number of T-cell lymphocytes is reduced, resulting in a poor lymphocyte-mediated immune response to the tumor.

Most studies have suggested that this preoperative score is a prognostic indicator in the early stages of different cancers (lung, kidney, breast, …) as well as in the late stages (Forget).

In breast cancer patients, the prognostic value of the preoperative NLR has been observed both for mastectomy and conservative breast surgery [20].

### 2.3. Statistical Analysis

In order to verify that the different groups were well balanced, a regression model was used. A Wilcoxon test was performed to ensure that there were no significant differences between the groups.

Data were analyzed using R-core Team Software, 2021 (url: http://www.r-project.org, accessed on 10 March 2022). *p* values < 0.05 were considered statistically significant [21]. Student’s test was used to compare the different scores: pain scores, distress scores and anxiety scores, and NLR and CRP values.

## 3. Results

Tumor and patient characteristics are summarized in Table 1.

Treatment modalities are described in Table 2.

We planned to include 300 patients. A total of 284 patients were finally included.

Four patients were excluded: three from the GAVRH group because the virtual reality session was interrupted due to headaches or anxiety generated by the program, and one patient in the GA group withdrew her informed consent.

Two hundred and eighty patients were finally included in our study, and we have complete results for 95 patients in the hypnosis sedation (HYPS) group, 93 patients in the general anesthesia (GA) group and 92 patients in general anesthesia + virtual relaxation (GAVRH) group.

For all mentioned patients, we have an evaluation of clinical and biological parameters.

### 3.1. Results of Pain Score

The results of pain scores observed among the three groups of patients show that acute pain was well controlled with analgesic support.

We observed lower pain scores on day 0, day 1 and day 8 in the HYPS group than in the other two groups.

The results of pain scores are detailed in Figure 1 and Table 3. They are expressed as confidence intervals in Table 3.

Because patients in the HYPS group had lower pain scores, the duration of NSAID use was statistically decreased in comparison with the two GA groups.

In the HYPS group, only two patients took anti-inflammatory drugs for more than 24 h after the surgical procedure. They are detailed in Figure 2.

### 3.2. Anxiety and Distress Scores

Anxiety and distress scores were high in all three groups at baseline (measures performed before the surgical procedure). Perfect agreement is observed between the measurements of the distress thermometer and the visual anxiety scale.

Breast cancer patients are known to be very stressed patients [22,23].

On day 1, the anxiety scale and distress score were statistically lower in the HYPS group than in the other two groups.

On day 8, the value was higher than on day 1 in the three groups, because this corresponds to the day of the postoperative consultation in which the pathological results and the proposal of a personal therapeutic plan were communicated to patients. However, despite this, the value was reduced in the HYPS group.

The value was lower in the GAVRH group in comparison with the GA group.

These values are detailed in Figure 3 and Table 4.

**Table 3 cancers-17-00049-t003:** Anxiety values with confidence intervals.

	GA Group N = 93	GAVRH Group N = 92	HYPS Group N = 95
Day 0	9.0 [8.91–9.09]	9.0 [8.91–9.09]	9.1 [9.00–9.20]
Day 1	7.4 [7.21–7.59]	5.4 [5.19–5.61]	4.0 [3.73–4.27]
Day 8	8.2 [8.00–8.40]	7.0 [6.77–7.23]	5.5 [5.22–5.78]

### 3.3. Results of Inflammatory Markers

The results of the NLR and CRP are noted in Table 5, Table 6 and Table 7 and Figure 4 and Figure 5. Inflammatory parameter results are expressed as confidence intervals in Table 5 and Table 6; *p*-values were calculated for the three groups at D0, 1 and 8, and detailed results can be found in Table 7.

The consumption of NSAIDs is illustrated in Figure 2.

The basal values of the CRP and NLR do not exhibit statistical differences among the three groups of patients.

The most significant differences were observed on day 1 between the HYPS group and the two groups of patients operated on while on general anesthesia with or without a virtual reality session. The *p* values for the comparison between the HYPS group and the GAVRH group and between the HYPS group and the GA group are 0.00009 and 0.00003 for the NLR. With regard to the CRP, a comparison between the HYPS group and the GAVRH group and the HYPS group and the GA group yields *p* values of 0.00006 and 0.00012.

On day 8, there were no statistical differences in biological markers among the three groups.

## 4. Discussion

This study has several limitations. First, it was a non-randomized study. For hypnosis sedation itself, it is not possible to propose a randomized design. The hypnotic process remains a choice of the patient and requires their motivation and active involvement.

Secondly, the number of patients was set at 300 patients and was not fully achieved, but with the arrival of the COVID-19 crisis, it was not possible to extend the study. Thirdly, the virtual reality program used, which consisted of an aquatic universe, proved suboptimal, and a number of patients were frightened by their aquatic environment. Fourthly, this study is unblinded because the types of anesthesia (general anesthesia and hypnosis sedation) are very different.

Although this study was not randomized, a large number of patients were included. The patients were prospectively included, and much data (clinical and biological markers) are available.

The three groups were well balanced in terms of the number of patients (no statistical difference between the three groups), and the groups were homogenous concerning age, menopausal status and histological subtypes.

In regard to the evaluation of the treatment modalities, the percentage of mastectomies was low and comparable between the HYPS group and the GA group, but this percentage was much higher in the GAVRH group.

In regard to the administration of radiotherapy and endocrine therapy, no statistical differences were observed between the three groups.

The percentage of chemotherapy was comparable between the GA group and the HYPS group, and the number of patients receiving chemotherapy was a little higher in the GAVRH group.

Breast cancer patients are known to be very anxious and stressed patients.

We confirm here that on day 0, patients in the three groups were indeed very stressed.

Analysis on day 1 confirms that hypnosis sedation and a virtual reality session were able to decrease distress and anxiety. These observations confirm the previous results of our study published in 2017 [10,12] and studies in which VRH was able to reduce perioperative anxiety [24,25].

On day 8, patients received their results; anxiety and distress scores were increased in the three groups, but the increase was lower in the HYP group.

Pain scores were low on day 0 among the three groups of patients. This means that the analgesic drug schedules used were efficient.

One of the major aims of analgesic drug schedules is to control acute pain and, by this means, to decrease the incidence of chronic pain syndrome [1,2,3,4,13,26,27,28].

Post-surgery pain control is essential for quality of life and remains a cornerstone to avoid chronic pain.

In this context, the observed values of pain scores are correct in our study and demonstrate the efficacy of the analgesic schedules.

We observed very good pain control in the HYPS group and a short duration of NSAID drugs consumption. A French study named HYPNOSEIN conducted by Amraoui and colleagues evaluated the effects of a hypnosis session before general anesthesia on postoperative outcomes in patients who underwent minor breast cancer surgery. No benefit of hypnosis on postoperative breast pain was observed. However, hypnosis seemed to have other benefits regarding fatigue, anxiety and patient satisfaction. A 15 min hypnosis session was planned, but the median duration reported is 6 min (2–15 min), which seems to be too short to generate an effect on postoperative pain. Another important difference from our study is the type of performed surgery: in our study, all patients underwent axillary sampling, sentinel node biopsy or axillary dissection, but in the study by Amraoui, surgery was limited to the breast [29]. The pre-emptive session of hypnosis was not a success in this study, contrary to Montgomery’s observations, who used hypnosis before breast surgery and confirmed a positive effect on pain [10]. The relative failure of the pre-emptive hypnosis in the study HYPNOSEIN should not condemn this type of use of hypnosis. It may be that the best option is to combine pre- and post-emptive sessions with the aim of achieving better pain control.

In a study performed by Laird [30], the authors examined the relationship between pain and systemic inflammation among a large group of cancer patients. They demonstrated that pain is correlated with systemic inflammation. The measure has been used with CRP in cancer patients: increasing levels of systemic inflammation were associated with worse symptoms, and patients reported worse outcomes. One of the explanations proposed by the authors is that pro-inflammatory cytokines cause pain by various means. The authors highlighted a correlation between CRP (as a surrogate marker of Interleukin 6 (IL6)) and pain.

In our study, we evaluated two inflammatory parameters (NLR and CRP) on day 0, day 1 and day 8.

In 1863, Virshow first proposed the role of inflammation after observing the presence of leucocytes in neoplastic tissue [31].

Since Virshow’s initial observations that inflammation and cancer are linked, empirical evidence has underscored inflammation as both a cause and a consequence of cancer [31,32,33,34,35].

Cancer-related inflammation promotes the neoplastic transformation of chronically inflamed tissue and sustains disease progression in developed tumors.

The continuous recalling of inflammatory signals in the tumor environment generates a progression of invasion [32,33,34,35,36].

Paradoxically, it is well recognized that cancer surgery (the removal of the primary tumor) can promote, by means of an inflammatory reaction, tumor metastasis (disseminating tumor cells or stimulating the growth of pre-existing micrometastases and worsening prognosis) [33,34,35,36,37,38,39,40,41,42,43,44].

Different means to decrease perioperative inflammation have been and are currently being studied, such as the consumption of NSAIDs and locoregional anesthesia [37,38,39,40,41,42,43,44,45,46].

Local anesthetic agents have some very interesting properties: [47,48] they modulate non-receptor protein kinases and voltage-gated sodium channels. The suppression of these pathways can inhibit cell proliferation and invasion, and facilitate focal adhesion, which could reduce metastatic risk. These data have been supported by some retrospective studies but not confirmed by others [45]. Recently, the results of a randomized study by Badwe suggested that the peritumoral infiltration of lidocaine at the time of surgery leads to statistically significant and clinically meaningful improvements in OS (overall survival) and DFS (disease-free survival) in patients with early-stage breast cancer [49].

To avoid the impact of the peritumoral infiltration of local anesthesia, it was administered in all patients in our study, and injection was performed according to the same modalities.

No difference was observed between the three groups in the NLR and CRP on day 0.

In a Belgian study [31], the mean value of the NLR of a person in good health was approximately 1.65.

In our study, among breast cancer patients, the basal values of the NLR were found to be higher in the three groups (between 2.3 and 2.5).

On day 1, NLR and CRP values were significantly lower in the HYPS group in comparison with the two other groups of patients operated on while on general anesthesia. This has already been demonstrated in a randomized study comparing general anesthesia and hypnosedation for endocrine surgery; in this study, IL6 values were also reduced in the hypnosedation group [50].

On day 8, no statistical difference was observed between the three groups.

One possible explanation for this difference may be the consumption of NSAIDs.

In the hypnosis sedation group, only two patients with inflammatory disease (polyarthritis) took NSAIDs for a prolonged period.

The use of NSAIDs can explain the fact that on day 8, no difference was observed between the three groups for this inflammatory parameter.

We did not observe any impact of the virtual reality session on biological parameters.

Clinical hypnosis, virtual reality and virtual reality hypnosis are three non-pharmacological approaches developed for pain management.

It is important to keep in mind that pain physiology is influenced by a complex network of interactions involving the autonomic, peripheral and central nervous systems as well as the endocrine and immune systems. Initial nociceptive stimulation is conducted along primary afferent fibers connected to the dorsal horn of the spinal cord. Then, the information ascends to the thalamus via the spinothalamic tract and to specific areas in the brainstem via the spinomesencephalic and spinoreticular tracts. From the thalamus, nociceptive information is projected to the cortex, where cognitive processing integrates sensory discriminative (location, intensity) and affective emotional (pain, unpleasantness) aspects. Research using positron emission tomography (PET) and functional magnetic resonance (fMRI) has unmasked a complex network of brain regions that are involved in the cognitive processing and modulation of pain.

Virtual reality hypnosis and hypnosis sedation have certain similarities and share common objectives but are not similar. To date, different hypotheses have been proposed regarding the pain attenuation effect of virtual reality hypnosis: the neurobiological interplay of brain cortices and neurochemistry as well as emotional and cognitive processes (attention and emotion). It is well known that hypnosis modifies painful stimulation by inducing a state of dissociation, while VR analgesia is based on distraction. A recent study performed by Rousseaux et al. [51] on healthy subjects confirms that VRH modulates cerebral pain processing and modified body physiology. Nonetheless, continuous research in an attempt to explain VRH analgesic mechanisms is still in progress and is warranted, because the mechanisms used by VRH to reduce pain have only just been decoded. In contrast, the mechanisms used by medical hypnosis and hypnosis sedation to reduce pain have been extensively studied. Many hypotheses have been formulated to explain pain relief induced by hypnosis. During hypnosis, several changes in brain functions occur in all the areas of the pain network and also in other brain areas. These different changes have been observed in neuroimaging studies [52].

Among the different hypotheses, there is a hypothesis called “the block of pain”, which occurs during hypnotic-focused analgesia, in response to a suggestion made during a hypnotic trance. Other complex mechanisms are also implicated, such as the cerebellar activation observed in functional magnetic resonance imaging, which might contribute to pain reduction by inhibiting the cerebral cortex. Other interactions with the endocrine and the immune systems must also be mentioned to explain the reduction in pain induced by hypnosis. Finally, the positive impact exerted by hypnosis on anxiety also plays a crucial role in pain relief.

In our study, we did not observe an impact on pain score, but the duration of the virtual reality session was short (only 20 min).

Three patients were excluded from the study because the virtual reality session was not tolerated and was interrupted due to discomfort generated by the virtual reality headset.

We also think that the type of program of virtual reality needs to be examined.

The length of the program is probably insufficient to generate a calm state.

The type of program needs to be diversified because immersion in the aquatic environment generates anxiety for some patients instead of calming them.

In this study, hypnosis sedation reveals its ability to significantly reduce the perioperative inflammatory process linked to the surgical procedure.

In the context of oncologic surgery, this reduction could be beneficial and decrease the risk of cancer progression [38,44,45].

## 5. Conclusions

Our study confirms our previous results, showing the positive effects of hypnosis sedation on distress, anxiety and pain, with a significant decrease in NSAID consumption [53,54,55].

This study also proposes a biological explanation for some of the benefits generated by hypnosis sedation.

Hypnosis sedation significantly decreases the early inflammatory reaction induced by the surgical procedure, both by itself and with the use of local anesthesia.

All the patients included in our study had early-stage breast cancer. This study was not randomized, and the number of patients was too small with a duration of follow-up too short to observe any impact on disease-free survival and overall survival.

Nevertheless, a reduction in the inflammatory process is always beneficial in reducing pain and the consumption of analgesic drugs [56].

Hypnosis sedation can be used as a novel strategy to limit the perioperative inflammatory process (linked to oncologic surgery), but to observe benefits—other than an improvement in quality of life—it is necessary to conduct new studies with a large number of patients and to follow them over a very long period.

## Figures and Tables

**Figure 1 cancers-17-00049-f001:**
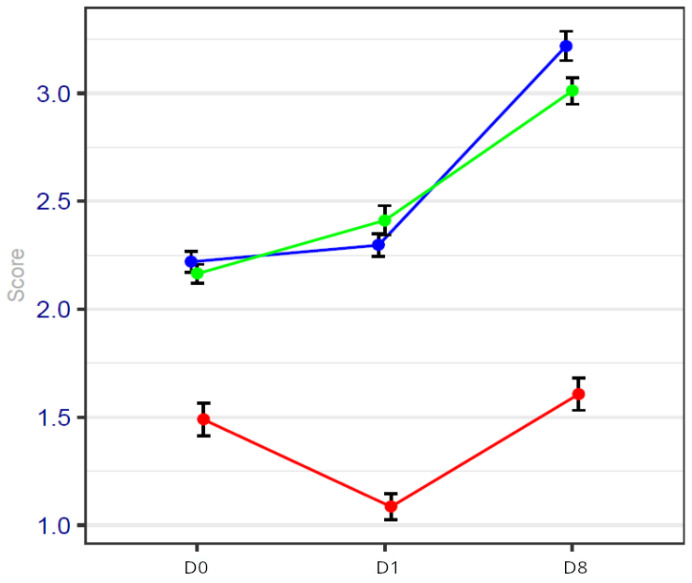
Pain score on days 0, 1 and 8 of the procedure. Blue line represents general anesthesia; green line represents general anesthesia + virtual reality; red line represents hypnosis sedation. The pain score is significantly reduced on day 1 in the HYPS group.

**Figure 2 cancers-17-00049-f002:**
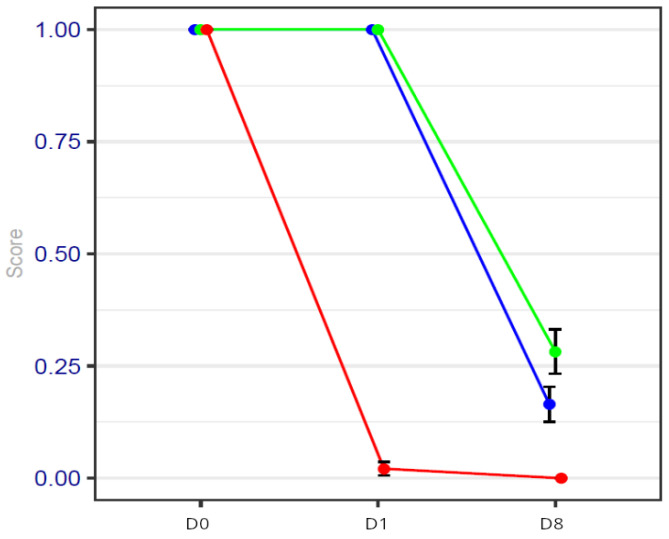
NSAID consumption values on days 0, 1 and 8 of the procedure. Blue line represents general anesthesia; green line represents general anesthesia + virtual reality; red line represents hypnosis sedation. The consumption of NSAIDs is significantly reduced on day 1 in the HYPS group.

**Figure 3 cancers-17-00049-f003:**
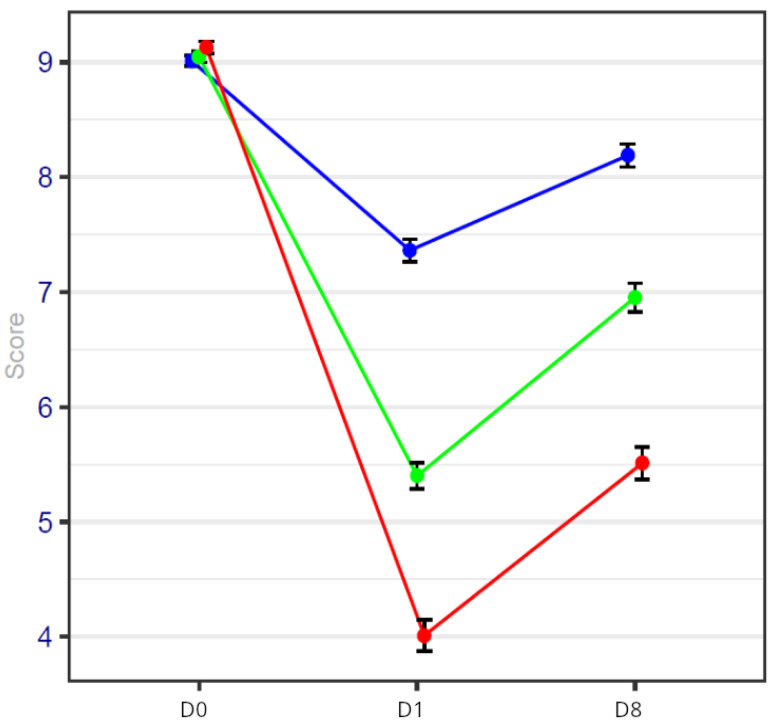
Anxiety score on days 0, 1 and 8 of the procedure. Blue line represents general anesthesia; green line represents general anesthesia + virtual reality; red line represents hypnosis sedation.

**Figure 4 cancers-17-00049-f004:**
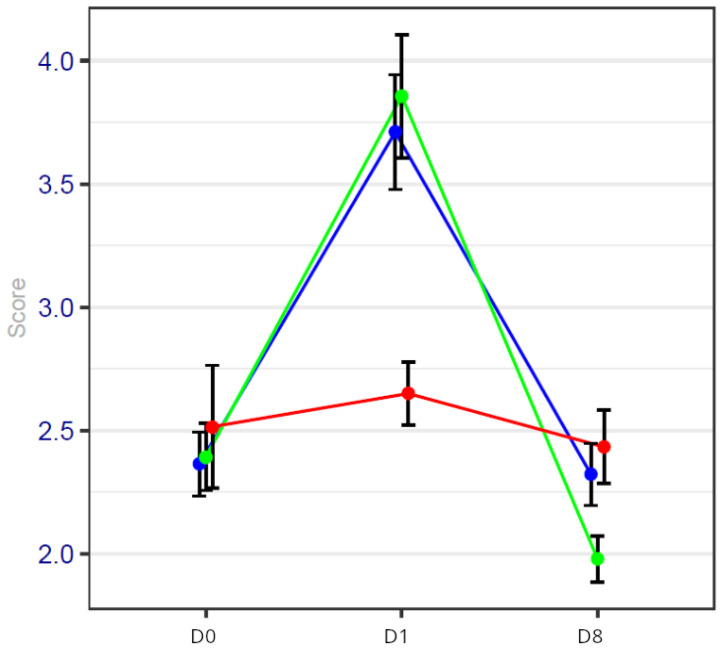
NLR values on days 0, 1 and 8 of the procedure. Blue line represents general anesthesia; green line represents general anesthesia + virtual reality; red line represents hypnosis sedation. NLR values are significantly reduced on day 1 in the HYPS group.

**Figure 5 cancers-17-00049-f005:**
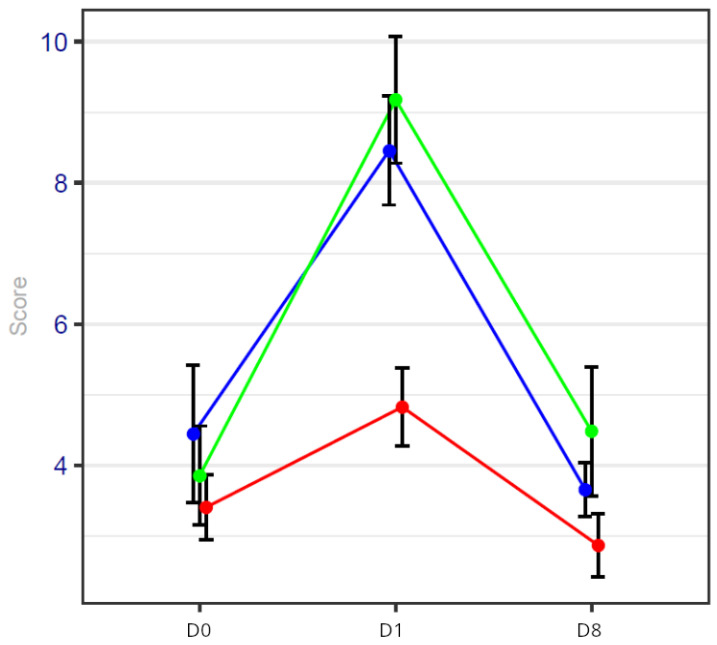
CRP values on days 0, 1 and 8 of the procedure. Blue line represents general anesthesia; green line represents general anesthesia + virtual reality; red line represents hypnosis sedation. CRP values are significantly reduced on day 1 in the HYPS group.

**Table 1 cancers-17-00049-t001:** Patient and tumor characteristics.

	HYPS Group (n = 95)	GA Group (n = 93)	GAVRH Group (n = 92)
Age (mean)	60.5 (34–86)	62 (28–83)	61 (34–86)
SD	12.16	12.13	11.5
Menopausal status			
- Pre	33	33	33
- Post	62	60	59
Histological subtypes			
- IDC	75	69	69
- ILC	13	13	18
- Pure grade III DCIS	7	11	5

IDC: invasive ductal carcinoma; ILC: invasive lobular carcinoma; DCIS: ductal carcinoma in situ.

**Table 2 cancers-17-00049-t002:** Treatment modalities.

	HYPS Group (n = 95)	GA Group (n = 93)	GAVR Group (n = 92)
Surgery			
- Lumpectomy			
+ SLNB	80	78	64
+ AD	3	3	8
- Mastectomy			
+ SLNB	6	7	16
+ AD	6	5	4
Chemotherapy	29	33	34
Anti-HER2 therapy (trastuzumab ± pertuzumab)	14	9	16
Radiotherapy	82	79	76
Endocrine therapy	76	71	76

SLNB: sentinel lymph node biopsy; AD: axillary dissection.

**Table 4 cancers-17-00049-t004:** Pain values with confidence intervals.

	GA Group N = 93	GAVRH Group N = 92	HYPS Group N = 95
Day 0	2.2 [2.10–2.30]	2.2 [2.12–2.28]	1.5 [1.35–1.65]
Day 1	2.3 [2.20–2.40]	2.4 [2.27–2.53]	1.1 [0.98–1.22]
Day 8	3.2 [3.06–3.33]	3.0 [2.88–3.12]	1.6 [1.46–1.74]

**Table 5 cancers-17-00049-t005:** NLR values are expressed with confidence intervals for the different periods in each group.

	GA Group N = 93	GAVRH Group N = 92	HYPS Group N = 95
Day 0	2.36 [2.11–2.61]	2.39 [2.16–2.62]	2.51 [2.04–2.98]
Day 1	3.71 [3.27–4.15]	3.85 [3.43–4.27]	2.65 [2.41–2.89]
Day 8	2.32 [2.08–2.56]	1.97 [1.81–2.13]	2.43 [2.15–2.71]

**Table 6 cancers-17-00049-t006:** CRP values are expressed with confidence intervals for the different periods in each group.

	GA Group N = 93	GAVRH Group N = 92	HYPS Group N = 95
Day 0	4.44 [2.55–6.33]	3.85 [2.53–5.17]	3.40 [2.51–4.29]
Day 1	8.45 [6.95–9.95]	9.17 [7.48–10.86]	4.82 [3.75–5.89]
Day 8	3.65 [2.91–4.39]	4.47 [2.75–6.19]	2.86 [1.99–3.73]

**Table 7 cancers-17-00049-t007:** Results of Student’s test and p-values for inflammatory parameters.

Groups Compared	Day 0	Day 1	Day 8
NLR—HYPS vs. GAVRH	0.44	0.00009	0.05
NLR—HYPS vs. GA	0.58	0.00003	0.44
CRP—HYPS vs. GAVRH	0.61	0.00006	0.23
CRP—HYPS vs. GA	0.33	0.00012	0.05

## Data Availability

The datasets generated during and/or analyzed during the current study are available from the corresponding author on reasonable request.

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
