# Peer review of "Hypnosis Sedation Used in Breast Oncologic Surgery Significantly Decreases Perioperative Inflammatory Reaction"

_cancers, 2024, doi:10.3390/cancers17010049_

Round 1

Reviewer 1 Report

Comments and Suggestions for Authors

Thank you for your submission.

In this study, hypnotic sedation was used as an anesthetic method for breast cancer surgery, reducing the use of painkillers and suppressing the increase in NLR on the first day after surgery. Based on this, it was concluded that hypnotic sedation suppresses the inflammatory response in breast cancer surgery, but the results of long-term prognosis have not yet been obtained. There are several things that need to be corrected.

First, American English and British English are mixed.

Line 20 contains '(' and is not grammatically correct.

The method of hypnotic sedation and virtual reality session needs to be explained.

The period is incorrectly inserted in line 148.

The text in table 5 is too small.

'[' is incorrectly inserted in line 257.

'( )' is used instead of '[ ]' in line 307.

Please check the period and lowercase letters in line 314.

There are many other grammatical errors. I think these things are factors that detract from your creative research.

Thank you.

Author Response

3. Point-by-point response to Comments and Suggestions for Authors

responses to Reviewer 1

Comments 1: American and British English are mixed

Response 1: American English has been suppressed

Comments 2: hypnotic sedation was used as an anesthetic method for breast cancer surgery, reducing the use of painkillers and suppressing the increase in NLR on the first day after surgery

Based on this it was concluded that hypnosis sedation suppresses the inflammatory response to breast cancer surgery but the results of long term prognosis have not yet been obtained. There are several things that need to be corrected

Response 2: Agree. I/We have, accordingly, /changed/

we fully agree with the reviewer’ comment :our study only shows that hypnosis sedation is able to decrease the perioperative inflammatory reaction (it is detailed in the texte)but we have no data -the follow up period is currently too short- demonstrating an impact of hypnosis sedation on desease free survival and /or overall survival. In addition, the statistical power of the study will be insufficient to obtain a significant result. We have insisted on this point and a phrase has been added in the conclusion page 17 (line 482)

3. comment3 the method of hypnotic sedation and virtual reality session need tobe explained

Response -the method of hypnosis sedation and virtual reality session have been described page 3-4-5 paragraph 2.1 line 87

4. Response to Comments on the Quality of English Language

Point 1:

Response 1: the quality of English language has been reviewed grammatical errors have been corrected ( suppressed, remplacement )

5. Additional clarifications

[parentheses have been suppressed, (line 20 page 1and other modifications performed in the text : title of table 5 modified (page 11) parentheses replaced by[],

Reviewer 2 Report

Comments and Suggestions for Authors

Dear authors,
reading the summary of your article, it is not possible to clearly understand that patients from the HYP group were not operated on under general anesthesia but only under hypnosis, so I suggest that you emphasize this in the Simple Summary as well as in the Abstract.

Author Response

Response to reviewer 2

comment :it is not possible to clearly understandthat patients from the HYP groip were not operatedon under general anesthesia

response:we thank the reviewer for his remarks and very positive comments

we have confirmed in the simple summary page 1(line 20) and in the abstract (line 32) that patients operated under hypnosis sedation were exclusively operated with hypnosis and not under general anaesthesia.

Reviewer 3 Report

Comments and Suggestions for Authors

In this study, Berliere et al. studied the effect of hypnosis sedation in breast oncologic surgery and found that it decreases perioperative inflammatory reaction and improves the postoperative pain condition.

My comments are here:

 1. Authors also used virtual reality hypnosis in comparison to sedative hypnosis techniques. Please elaborate in the discussion the differences and mechanisms of the duo.

 2. Please increase the quality of the graphs and put significant * on graphs to show the difference among the groups. 

 3. Please explain the hypnosis protocol in detail in the methods section.

 4. A previous study did not find any relief in postoperative pain using hypnosis (PMCID: PMC6324272). Please compare it in the discussion part.

 5. Which one is a better approach: Preemptive or postemptive hypnosis, or both, to relieve the patients' pain more significantly? Add in discussion.

 6. What is the mechanism of analgesia of hypnosis in patients? Is it directly alleviating the pain pathways through immune regulation or via other pathways (relieving anxiety associated with surgery, by activating the ANS etc?). Please explain in discussion section.

 7. Lines #207, #212: Where are the figures 1c, 1d and 1e? Please edit the figure numbers in entire paper and match the etxt with the original figure numbering.

 8. Please clearly highlight the limitations (less ‘n’ number, lack of randomization, blinding, etc.) in different sections or in the Discussion.

Comments on the Quality of English Language

It has some typos and the overall tone may be improved.

Author Response

Responses to reviewer 3

comment1 -1 authors also used virtual realityhypnosis in comparison tosedative hypnosis. Please elaborate in the discussion the differences and mechanisms of the duo

response:differences and mechanisms of hypnosis sedation and virtual reality have been added in the discussion pages 15 and 16 (discussion line 417) references added 51-52 page 21-22

Comment 2:Please increase the quality of the graphs

Response we have tried to increase the quality of the graphs and have added significant results .

-Comment 3 please explain the hypnosis protocol in detail in the methods section

Response the hypnosis protocol and the virtual reality session have been explained pages 3-4-5 (paragraph 2.1 added line87

comment 4 :a previous study did not find any relief in postoerative pain using hypnosis

response :-4 the results of the HYPNOSEIN study have been added and commented in the discussion Page 14(line 346). Reference 29 has been added

Comment 5: which one is a better approach: preemptive or postemptive hypnosis, or both , to relieve the patients’pain more significantly

Response:-5 preemptive and postemptive hypnosis have been discussed (line 355) page 11

Comment 6 : what is the mechanism of analgesia of hypnosis in patients please explain in the discussion section

response mechanisms of analgesia induced by hypnosis have been discussed. Pages 15 and 16(discussion line 447)

comment 7

Where are fig 1c and and 1e

response numerotation of figures has been corrected (12-3-4-5 )and figures have been added.

Comment 8 please clearly hilight the limitations in different sections or in the discussion

Response limitations of the study have been added and detailed in the discussion page 13 line 305

Round 2

Reviewer 1 Report

Comments and Suggestions for Authors

I have confirmed that many parts have been revised well. I don't know if NRS was incorrectly written as 'numerical pain waiting score' in line 172, or if it is a new score called 'numerical pain waiting score'. 

If only that is revised, I think it is okay to accept.

Author Response

I  Would like to thank reviewer 1 for taking the time to review our manuscript and for all his comments which improved the quality of our manuscript. I have corrected the error on line 172: numerical pain rating score (NRS).

Reviewer 3 Report

Comments and Suggestions for Authors

I am satisfied with the changes done by the authors and the current manuscript seems improved than the original version.

Author Response

I  would like to thank reviewer 3 for taking time to proofread our manuscript and for all his - her comments  which helped to improve the quality of the manuscript